# Histopathological Alterations in *Nilaparvata lugens* (Hemiptera: Delphacidae) after Exposure to *Cordyceps javanica*

**DOI:** 10.3390/insects15080565

**Published:** 2024-07-26

**Authors:** Peerasak Bunsap, Sinlapachai Senarat, Seree Niyomdecha, Chaninun Pornsuriya, Gen Kaneko, Narit Thaochan

**Affiliations:** 1Agricultural Innovation and Management Division, Faculty of Natural Resources, Prince of Songkla University, Songkhla 90110, Thailand; peerasakbunsap@gmail.com (P.B.); chaninun.p@psu.ac.th (C.P.); 2Center of Excellence on Agricultural Biotechnology (AG-BIO/MHESI), Bangkok 10900, Thailand; 3Division of Biological Sciences, Faculty of Science, Prince of Songkla University, Songkhla 90110, Thailand; sinlapachai.s@psu.ac.th (S.S.); saree.n@psu.ac.th (S.N.); 4College of Natural and Applied Science, University of Houston-Victoria, Victoria, TX 77901, USA; kanekog@uhv.edu

**Keywords:** entomopathogenic fungi, histopathology, the brown plant hopper, Thailand

## Abstract

**Simple Summary:**

We exposed brown planthoppers (BPHs) *Nilaparvata lugens* to the entomopathogenic fungus *Cordyceps javanica* PSUC002 and observed morpho-histological changes from 0 to 120 h post-inoculation (pi). At 12 h pi, we observed the first filamentous fungi on the external integument and the first fungal conidia penetrated host planthopper cells at 24 h pi. In contrast, we observed the initial degeneration of BPHs at 6 h pi, especially in the integument and adipose tissue. We also found the degeneration of integument and adipose tissue at 12 h pi, and their completed necrosis became clear at 96 h pi. This study illustrated the process of *C. javanica* infection in BPHs and demonstrated that fungal agents can be used to control the BPHs in an integrated pest management (IPM) program.

**Abstract:**

The brown planthopper (BPH), *Nilaparvata lugens* (Stål, 1854)*,* is a pest of rice plants worldwide. *Cordyceps javanica* is a destructive entomopathogenic fungus known to attack leafhoppers or BPHs specifically. Live adult BPH samples were inoculated with isolated *C. javanica* PSUC002, and their interaction was morpho-histologically examined from 0 to 120 h post-inoculation (pi). We observed that the mortality of BPH continuously increased until 120 h pi (Day 5). Tissue alterations in the host were examined after infection using morphological and histological methods, including the Grocott Methenamine Silver stain test (GMS). Filamentous fungi were first found on the external integument at 12 h pi, and fungal conidia attached to the integument at 24 h pi. However, the initial degeneration of BPHs was identified by histology at 6 h pi especially in the integument and adipose tissue. We identified the degeneration and loss of integument and adipose tissue of infected BPHs at 12 h pi, and their necrosis was completed at 96 h pi. The enzymatic index of the sampled fungi (chitinase and protease) peaked at 7 days of incubation. This study demonstrated that *C. javanica* PSUC002 is useful to control the BPHs as an eco-friendly practice and will possibly be applied in agriculture.

## 1. Introduction

The brown planthopper (BPH), *Nilaparvata lugens* (Stål, 1854) (Hemiptera: Delphacidae), is one of the most serious primary pests in all rice-growing regions of the world [1]. The BPH is a monophagous phloem sap-sucking insect that stunts plant growth, turning the leaves yellow and dry, a state known as “hopper burn” [2]. The BPH also increases infections with the rice grassy stunt virus (RGSV) and rice ragged stunt virus (RRSV), causing unfavorable growth and rice harvest failure [1]. The use of chemical insecticides has helped farmers to control a variety of pests and to increase productivity [3] but has negatively affected the health of farmers and caused the problems of non-target effects and insecticide resistance [4,5]. The resistance of *N. lugens* to insecticides has been reported [6], necessitating the development of new and effective methods of controlling this pest.

An integrated pest management (IPM) strategy uses a combination of knowledge about pest control, safety procedures, and sustainable agricultural production [7,8]. Biological pest control with entomopathogenic fungi (EPFs) fits the concept of IPM and has become one of the most popular and effective non-chemical pest control methods [7,8]. Commonly used EPF species include *Cordyceps* spp., especially *Cordyceps fumosorosea* (Hypocreales: Cordycipitaceae), formerly *Isaria fumosorosea* Wize, 1904, which has been used to control sap-sucking insects such as *Jacobiasca formosana* (Paoli, 1932) (Hemiptera: Cicadellidae); *Aphis gossypii* Glover, 187 (Hemiptera: Aphididae); *Bemisia tabaci* Gennadius, 1889 Gennadius (Hemiptera: Aleyrodidae) [8,9]; and *Stephanitis nashi* Esaki & Takeya, 1931 (Hemiptera: Tingidae) [10]. Furthermore, these EPF species potentially play a role in plant–microbial symbiosis, in which they protect plants against various insects and diseases [11]. The cuticle degrading enzymes (CDEs) of EPFs are used as biological control agents against insects [12,13]. CDEs are known to have proteolytic, chitinolytic, and lipolytic activities [12,13,14].

Histopathological investigation is a critical source of primary data that can be used to assess tissue alterations and cellular abnormalities [15,16] and has unveiled the process of insect mortality following infection by parasitic fungi [17,18]. For example, Chen et al. [19] found *C. sorosea* spores in aphids at 36 h post-infection (pi), and its mycelium rapidly spread between 48 and 72 h pi. Tian et al. [20] reported that *C. fumosorosea* hyphae were formed within 6 h of infection by *B. tabaci* and continuously increased for 120 h. These findings shed light on the histological dynamics of fungal invasion and host response, which is quite diverse and specific to the EPF and host. It is thus necessary to have the spaciotemporal visualization of the infection process on each host and EPF combination, but there has been no histopathological assessment of *N. lugens* infected by EPFs. This study seeks to enhance our understanding of the tissue response of BPH to an EPF, potentially contributing to agricultural pest control in the future.

In this research, we observed the mortality rate and tissue alterations of *N. lugens* infected by *C. javanica* PSUC002. The fungus was identified by using a molecular approach, DNA sequencing followed by phylogenetic analysis, because it was difficult to identify the species of the field-collected fungi from external morphology. The activity of cuticle-degrading enzymes was also assessed.

## 2. Materials and Methods

### 2.1. DNA Extraction and Phylogenetic Analysis of Cordyceps javanica

We isolated the fungus *C. javanica* PSUC002 from an infected caterpillar in a plant conservation area of Khor Hong mountain at the Prince of Songkla University, Songkhla, Thailand (7.0163, 100.5197). This particular isolate was selected due to its demonstrated virulence observed in a laboratory [21].

*Cordyceps javanica* PSUC002 was cultured for three days on a Sabouraud dextrose agar with yeast extract (SDAY) medium (1% glucose, 0.25% yeast extract, 0.25% peptone, and 2% agar) to obtain young mycelia. DNA was extracted from the young mycelia using a mini-preparation method [21]. The presence of total DNA was confirmed by 1% agarose gel electrophoresis. PCR amplification was carried out in a thermal cycler from Bio-Rad Laboratories (Hercules, CA, USA). The internal transcribed spacer (ITS) region was specifically amplified using the universal primers ITS4: 5′-TCCTCCGCTTATTGATATGC-3′, and ITS5: 5′-GGAAGTAAAAGTCGTAACAAGG-3′ [22,23,24]. The PCR mixture consisted of a DNA template, 20 pmol of each primer, 2× OneTaq^®^ PCR master mix with standard buffer (Biolabs, New England, MA, USA), and nuclease-free distilled water. The PCR amplification involved an initial denaturation step at 94 °C for 30 s, followed by 30 cycles of denaturation at 94 °C for 30 s, annealing at 60 °C for 60 s, and extension at 68 °C for 1 min. At the end of the cycle, the final extension step at 68 °C was extended to 5 min. The resulting PCR product was stained with Novel Juice (GeneDirex, Taoyuan, Taiwan) and visualized using 1% agarose gel electrophoresis.

The PCR products were sequenced at Macrogen in Seoul, South Korea. The DNA sequence of the ITS region was then queried against databases using BLASTtn (National Center for Biotechnology Information, NCBI). To construct the phylogenetic tree, DNA sequences from the fungal isolate and related species were obtained from the GenBank database.

The obtained sequences were aligned with known sequences from the National Center for Biotechnology Information database using Bioedit v. 7.2 [25] and manually edited as necessary using MEGA X [26]. We constructed phylogenetic trees for each alignment using the maximum likelihood (ML), maximum parsimony (MP), and Bayesian inference (BI) methods. The ML tree was constructed using MEGA X based on the TN92 + G evolutionary model. The MP tree was constructed using the heuristic search option of MEGA X with 1000 random additions of sequences and branch-swapping by tree bisection and reconnection (TBR). The Bayesian tree was generated using MrBayes ver. 3.2.7a [27]. Markov chain Monte Carlo (MCMC) runs were performed for 1,000,000 generations and sampled every 100th generation. The initial 1000 generations were discarded as burn-in, with the remaining trees being used to calculate the Bayesian inference posterior probability values. Phylogenetic trees were visualized using FigTree ver. 1.4.4 (http://tree.bio.ed.ac.uk/software/figtree/ accessed on 1 March 2023).

### 2.2. Nilaparvata Lugens Preparation

The BPH adult specimens (3.5–4.0 mm) were from Phatthalung Rice Research Center Phatthalung Province, Phatthalung City, Thailand (Figure 1A–C; 7.5663, 100.1258). All samples were reared in a box cage (30 cm × 30 cm × 30 cm.) at 27 ± 1 °C with rice plants (28–30 days old) of strain TN1 (Taichung Native 1 from Phatthalung Rice Research Center).

### 2.3. Cordyceps javanica Preparation

Before molecular identification, this fungus was identified as *C. fumosorosea*, and the strain number PSUIsa001 was assigned to the sample. The fungus was cultured on SDAY medium, and the conidia were harvested from SDAY in 0.05% Tween-80 in distilled deionized water (ddH_2_O). Conidial suspension concentrations were determined using a Neubauer haemocytometer and were adjusted to 1.0 × 10^5^, 1.0 × 10^6^ 1.0 × 10^7^, and 1.0 × 10^8^ conidia/mL to test the mortality rate of BPH.

### 2.4. Infection and Morpho-Histological Examination

A conidial suspension of 1.0 × 10^8^ conidia/mL of *C. javanica* was sprayed on 200 BPH individuals on plastic plates. The sprayed specimens were kept at 27 ± 1 °C. Ten live and 10 dead BPHs were randomly collected at 0, 6, 12, 24, 36, 48, 60, 72, 84, 96, 108, and 120 h pi. The external morphology of specimens was examined at each time point and photographed with a Leica S8 APO stereo light microscope with an apochromatic 8:1 zoom lens (Histocenter, Germany). All BPHs were fixed in Bouin’s solution for 36–48 h and then stored in 70% *v*/*v* ethyl alcohol for further observation.

All fixed samples were dehydrated sequentially with an ethyl alcohol series, cleared with xylene, and embedded with paraplast. The paraffin blocks were sectioned at 4 μm using a rotary microtome (Bexco Exports, Ambala, India). Sections were stained with Harris’s hematoxylin and eosin (H&E) to investigate histological structures [28] and Gomori methenamine silver (GMS) to identify the presence of fungi [29]. All histological slides (three representative slides per sample) were examined both structurally and histopathologically under a compound light microscope (Leica ICC 50 W, Histocenter, Wetzlar, Germany). The fungal area in BPH samples was determined using the ImageJ 1.53 K application (NIH, Bethesda, MD, USA; available at https://imagej.nih.gov/ij/download.html, 1 March 2023), following [30,31].

### 2.5. Production and Measurement of Cuticle Degrading Enzymes

#### 2.5.1. Chitinase Activity

To evaluate the chitinase activity of *C. javanica* PSUC002, 5 mm mycelial agar discs (n = 3) were transferred onto modified SDAY + 1% colloidal chitin + bromocresol purple [32] and incubated at 25 °C for 5, 7, and 9 days.

#### 2.5.2. Protease Activity

To evaluate the protease activity of *C. javanica* PSUC002, 5 mm mycelial agar discs (n = 3) were transferred onto SDAY + 3% skim milk [25] and incubated at 25 °C for 5, 7, and 9 days. The enzymatic index (EI) was calculated using the following formula [33]: EI = Diameter of degradation zone (R)/Diameter of colony (r).

### 2.6. Data Analysis and Statistical Analysis

BPH mortality was corrected using Abbott’s formula and represented as a percentage (%) [30]. The mortality, fungal area, and enzyme index were expressed as means ± SE. All parameters were compared by ANOVA followed by Tukey’s HSD test. All data were analyzed by IBM SPSS Statistics 26.0 with a significance level of 95%.

## 3. Results

### 3.1. Phylogenetic Analysis

The DNA sequencing determined a 1814 bp nucleotide sequence of the rDNA-ITS gene from the fungal isolate. While the fungal isolate was originally thought to be *C. fumosorosea*, a BLAST search against GenBank databases revealed that the ITS gene sequence was 100% identical to *C. javanica* (GenBank accession number CBS134.22 and CBS174.25). The ITS was deposited in GenBank as LC822808. We therefore designated the fungal isolate as *C. javanica* PSUC002. Phylogenetic analysis showed that the *C. javanica* PSUC002 isolate was clustered with *C. javanica*, being clearly separated from the *C. fumosorosea* clade (Figure 2).

### 3.2. The Mortality Rate of BPH-Infected Cordyceps javanica

The percentage mortality (means ± SE) of BPHs exposed to *C. javanica* PSUC002 suspensions at different concentrations is shown in Figure 3. The mortality quickly increased at all concentrations until 120 h pi (Day 5). At 72 h pi (Day 3), a cumulative mortality of 84% was observed at concentrations of 1.0 × 10^7^ and 1.0 × 10^8^ conidia/mL, whereas only 44% mortality was observed at the concentration of 1.0 × 10^6^ conidia/mL. Exposure to 1.0 × 10^7^ and 1.0 × 10^8^ conidia/mL resulted in 100% mortality at 120 h pi.

### 3.3. Morphology and Histopathology of Nilaparvata lugens (BPH)

We examined the morphology and histopathology of BPH exposed to *C. javanica* PSUC002 at 1.0 × 10^8^ conidia/mL from 0 to 120 h pi (Figure 4 and Figure 5). At time 0, BPH showed no fungal spores or signs of histopathological alteration in external morphology nor in representative structures including the integument, adipose tissue, and muscles along the body (Figure 4A–C). No external morphological signs of infection were found at 6 h pi. However, histological staining found a few clusters of fungal spores penetrating the adipose tissue (Figure 4D–F). Hyphal growth significantly increased from 12 to 120 h pi (Figure 4G and Figure 5) as clearly indicated by the quantification of fungal areas in the tissue (Figure 6). At 60 h pi, the fungus exerted destructive impacts on BPH as represented by the necrosis of adipose tissue observed in all samples (Figure 5A–F). These lesions increased between 72 and 84 h pi (Figure 5G–L). The infected BPH sections at 96 h pi showed extensive hyphal growth and numerous fungal spores (Figure 5M–O). At 108 to 120 h pi, the entire body was completely covered by the mycelium of *C. javanica* PSUC002 and showed severe necrosis (Figure 5F–O).

### 3.4. Activity of Cuticle- and Protein-Degrading Enzymes

The chitinase and protease activities of *C. javanica* PSUC002 were measured semi-quantitatively (Figure 7, Table 1). The EI of chitinase activity varied between 1.00 and 2.77, and the EI of protease activity between 1.08 and 1.14. The maximum activity was recorded as 5 days for chitinase and 9 days for protease. It was observed that chitinase activity was more prominent than protease activity (Table 1). The chitinase activity also ceased to be detectable after 9 days of incubation (Table 1). Additionally, the enzyme indexes of all enzymes were significantly different (*p* < 0.05, Table 1).

## 4. Discussion

In this study, we aimed to investigate the tissue alteration of the BPH infected by *C. javanica* PSUC002 for the first time. *C. javanica* is an entomopathogenic fungus (EPF), and the obtained results can be used to assess its effectiveness for the biocontrol of the BPH. Several EPFs have been used to control insects including lepidopterous larvae, aphids, and thrips [34], and *C. fumosorosea* is one of the most important EPFs [8]. EPFs including *Beauveria bassiana* and *Metarhizium anisopliae* [35] have been used to control the BPH [10,36]. Indeed, previous observations have shown that more than 60% of insects have pathogenic microorganisms [37].

In this study, we used field-collected EPFs. The conidial characterizations of the anamorphic *Cordyceps* (formerly *Isaria*) are very similar, and hence they are known by a generic name instead of a clear scientific name [24,38]. Thus, DNA barcoding was the only reliable method to correctly identify the species of this fungal group. We used DNA analysis of a wild fungus isolated from an insect in the field. The isolate was shown to be a strain of *C. javanica*, which we named PSU002. While *C. fumosorosea* is a well-studied EPF, *C. javanica* has been successfully commercialized for controlling insect pests [39,40]. For example, an isolate *C. javanica* BE01 can cause high mortality in *Hyphantria cunea* within 4 days after inoculation [41]. Moreover, *C. javanica* IJ-tg19 showed mucilage secretion on the body of the aphid *Acyrthosiphon pisum* within 48 h post-inoculation [38].

In the present study, our morphological and histopathological analyses demonstrated that *C. javanica* PSUC002 penetrated the integument and reached the adipose tissue within 6 h pi and 12 pi throughout the 108 pi (summarized in Figure 8). Similarly, in *Bemisia tabaci* infected by *C. fumosorosea,* hyphae formation was observed within 6 h and increased through to 120 h pi [20]. However, the hyphal expansion in our observation was faster than that observed by Chen et al. [19], where *C. javanica* PSUC002 spores were observed in the body of aphids at 36 h pi and its mycelium had spread extensively at 48 and 72 h pi. Previous reports of fungal accumulations in beetle cuticles similarly noted hyphae attached to all body regions of the beetles [19,20]. However, the efficiency of fungal infection must be associated with insect species and environmental conditions including UV light, temperature, and humidity [34].

We also found that alterations of the integument and adipose tissue took place at 12 h pi in BPH, whereas the necrosis of tissue occurred at 60 h pi. Interestingly, a clear increase in BPH mortality was observed on Day 2, which was earlier than the occurrence of the necrosis. The fungal structures distributed within the body of BPH likely led to the death of the insect before inducing necrosis. The fungi likely penetrated the integument using extracellular cuticle-degrading enzymes, which include proteolytic and chitinolytic enzymes [12,13,14]. The enzymatic invasion to the body might be the cause of the mortality. The present study detected the activity of these enzymes in *C. javanica* for the first time (Table 1). The mechanistic pressure of specialized hyphal structures has also been reported in *Metarhizium* sp. and *B. bassiana* [42].

Protease enzymes released from EPFs play a crucial role in degrading the insect cuticle [12,13,14]. The activity of the chitinase and protease sampled from *C. javanica* peaked at different days of incubation, which has an implication on the time course of killing BPHs. Although the precise enzymatic strategies of how *C. javanica* PSU002 kills the host are not well known, our results suggest that it is a week-long process. The production of EPF extracellular enzymes has virulence potential for the pest insect [43]. The detailed time course should be investigated for the practical application.

## 5. Conclusions

Our data indicated that *C. javanica* PSUC002 can be used as a biological control agent against BPH as part of an integrated pest management (IPM) program as well as an eco-friendly evaluation, further strengthening the potential for its commercial use in agriculture.

## Figures and Tables

**Figure 1 insects-15-00565-f001:**
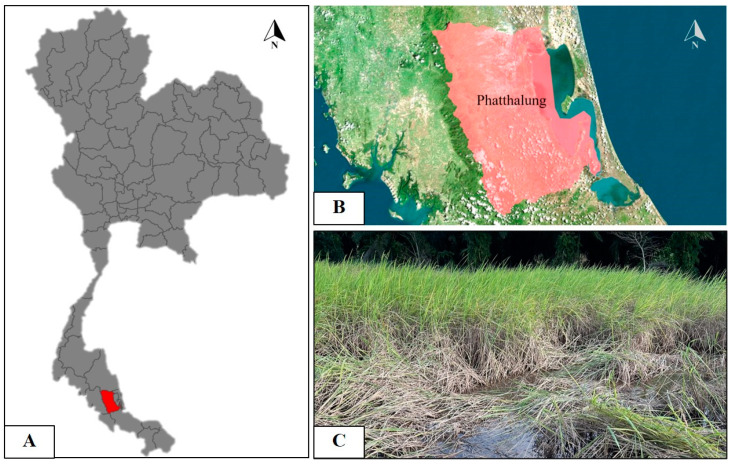
Map of locations showing the outbreaks of *Nilaparvata lugens* in Phatthalung province, Thailand (**A**,**B**). This insect is widely distributed in the rice field (**C**).

**Figure 2 insects-15-00565-f002:**
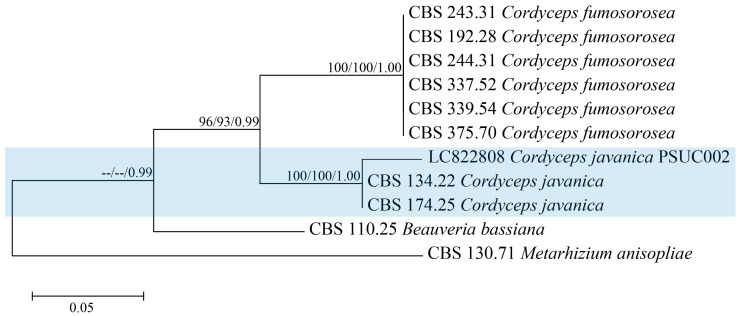
The phylogenetic tree was generated from maximum likelihood (ML) based on the combined ITS sequence data of *Cordyceps* spp. in *C. fumosorosea* and *C. javanica* (the blue highlight). Bootstrap values equal to or greater than 70% are indicated for maximum likelihood (ML) and maximum parsimony (MP) and equal to or greater than 0.95 for Bayesian posterior probabilities (BYPPs). The tree is rooted to *Beauveria bassiana* and *Metarhizium anisopliae*.

**Figure 3 insects-15-00565-f003:**
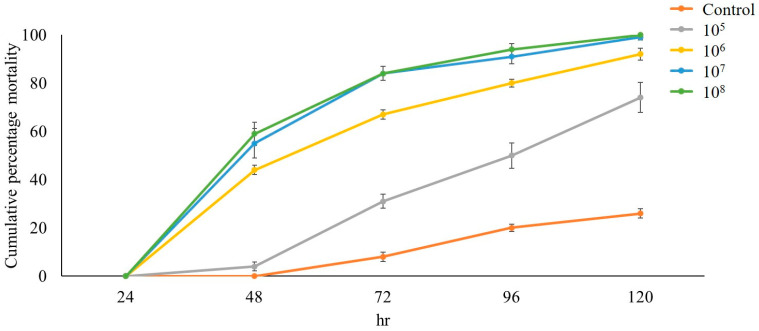
The cumulative percent mortality of *Nilaparvata lugens* after exposure to conidial suspensions of *Cordyceps javanica* PSUC002 at different concentrations (conidia/mL).

**Figure 4 insects-15-00565-f004:**
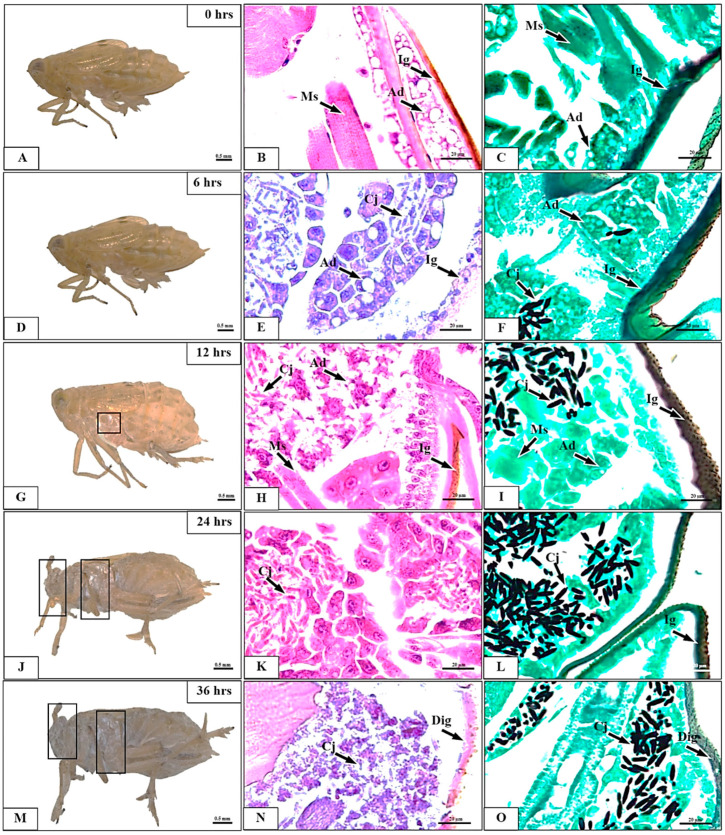
External morphology and histological changes in *Nilaparvata lugens* adults after exposure to a conidial suspension of *Cordyceps javanica* PSUC002 at a concentration of 1.0 × 10^8^ conidia/mL. (**A**–**C**) Normal appearance at 0 h. (**D**–**F**) The external appearance appeared normal without *C. javanica* at 6 h post-inoculation, but a few *C. javanica* had infiltrated the adipose tissue. The formation of *C. javanica* in the sampled insect tissue was significantly increased at 12 h (**G**–**I**), 24 h (**J**–**L**), and 36 h (**M**–**O**) post-exposure. Ig = integument, Ad = adipose tissue, Ms = muscle, Cj = *Cordycep javanica*, Dig = degeneration of integument. Staining methods: (**B**,**E**,**H**,**K**,**N**) = Harris’s hematoxylin and eosin (H&E); (**C**,**F**,**I**,**L**,**O**) = Gomori methenamine silver (GMS).

**Figure 5 insects-15-00565-f005:**
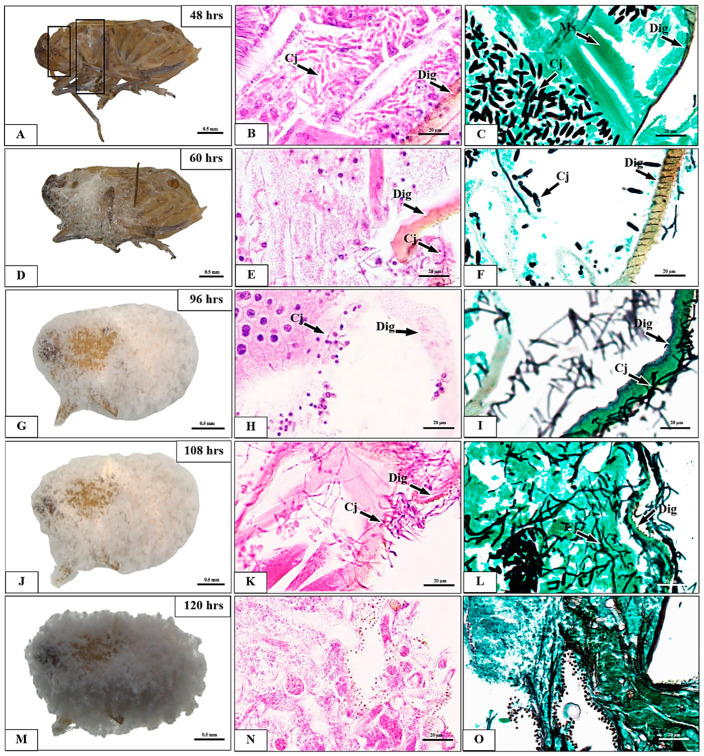
External morphology and histological changes in *Nilaparvata lugens* adults after exposure to a conidial suspension of *Cordyceps javanica* PSUC002 at a concentration of 1.0 × 10^8^ conidia/mL. (**A**–**C**) The formation of *C. javanica* on the body of the insect continues the degeneration of the integument (Dig). The conidia form hyphal bodies inside *N. lugens* which develop on the body from 48 to 120 h post-exposure (**D**–**O**). Ig = integument, Ad = adipose tissue, Ms = muscle, Cj = *Cordycep javanica*, Dig = degeneration of integument. Staining methods: (**B**,**E**,**H**,**K**,**N**) = Harris’s hematoxylin and eosin (H&E); (**C**,**F**,**I**,**L**,**O**) = Gomori methenamine silver (GMS).

**Figure 6 insects-15-00565-f006:**
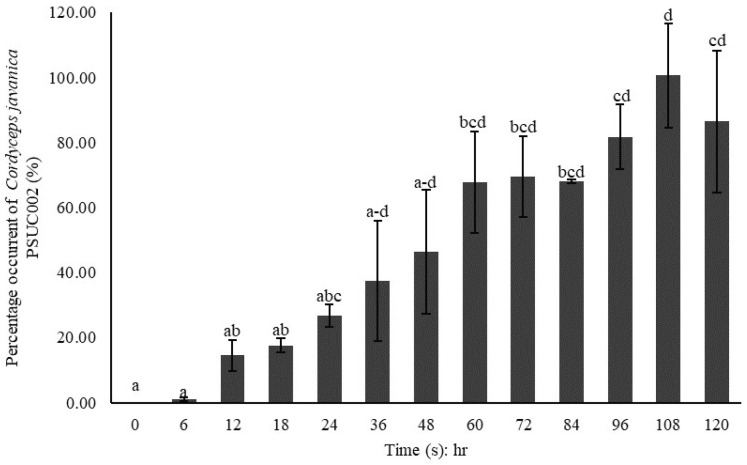
Expansion of *Cordyceps javanica* PSUC002 in the tissue of *Nilaparvata lugens* (means ± SE) from 0 to 120 days post-exposure. Different letters indicate a significant difference at *p* < 0.05.

**Figure 7 insects-15-00565-f007:**
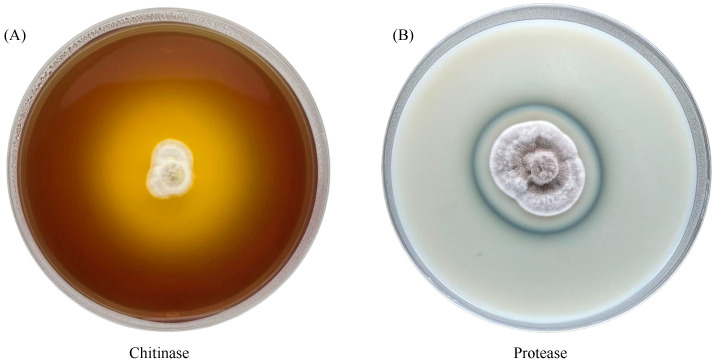
The chitinase (**A**) and protease (**B**) activity of *Cordyceps javanica* PSUC002 was investigated on SDAY + 1% colloidal chitin + bromocresol purple and SDAY + 3% skim milk media, respectively. The photographs show production after 5 days of incubation at 25 °C.

**Figure 8 insects-15-00565-f008:**
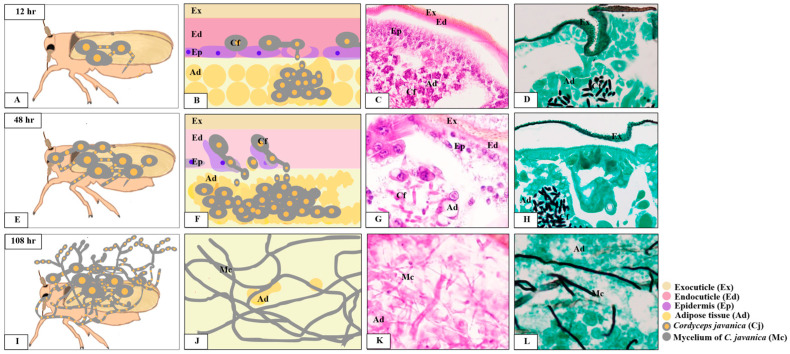
Diagrams show the progression of histopathological alterations in *Nilaparvata lugens* at 12, 48, and 108 h after exposure to *Cordyceps javanica* PSUC002. (**A**–**D**): A few clusters of *C. javanica* PSUC002 spores started to germinate in the sampled insect at 12 h post-exposure. (**E**–**H**): Prominent spores of *C. javanica* PSUC002 had germinated at 48 h and were distributed among adipose tissue. (**I**–**L**): The obvious features of fungal hyphae and mycelium of *C. javanica* PSUC002 developed among necrotic areas.

**Table 1 insects-15-00565-t001:** Enzyme index (EI) of the chitinase and protease produced by *Cordyceps javanica* PSUC002.

Enzymes	Diameter (mm) of Degradation Zone (R) (Diameter of Clear Zone + Diameter of Colony)			Diameter (mm) of Colony (r)			Enzyme Index (EI) = Diameter of Degradation Zone (R)/Diameter of Colony (r)		
	5 Days	7 Days	9 Days	5 Days	7 Days	9 Days	5 Days	7 Days	9 Days
Chitinase	35.00 ± 0.00 ^b^	53.33 ± 1.67 ^c^	24.17 ± 0.44 ^a^	12.67 ± 0.33 ^a^	20.00 ± 0.29 ^b^	24.17 ± 0.44 ^c^	2.77 ± 0.07 ^b^	2.67 ± 0.12 ^b^	1.00 ± 0.00 ^a^
Protease	19.50 ± 0.29 ^a^	30.83 ± 0.60 ^b^	42.00 ± 0.76 ^c^	17.67 ± 0.33 ^a^	28.67 ± 0.44 ^b^	36.83 ± 0.17 ^c^	1.10 ± 0.01 ^ab^	1.08 ± 0.01 ^a^	1.14 ± 0.02 ^b^

Values are expressed as mean ± SE. Different letters in the same row for each parameter indicate a significant difference at *p* < 0.05.

## Data Availability

The DNA sequence data obtained from this study were deposited in GenBank under accession numbers ITS (LC822808).

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
