# Peer review of "Histopathological Alterations in *Nilaparvata lugens* (Hemiptera: Delphacidae) after Exposure to *Cordyceps javanica"

_insects, 2024, doi:10.3390/insects15080565_

Round 1

Reviewer 1 Report

Comments and Suggestions for Authors

My suggestions are attached

Comments on the Quality of English Language

Author Response

Reviewer 1

Minor concerns:

Question 1- Line 15: were exposed to Cordyceps javanica

Answer: We agreed (Line 15).

Question 2- Line 16: post-inoculation

Answer: We agreed (Line 16).

Question 3- Line 17: In contrast, the initial degeneration

Answer: We agreed (Line 17).

Question 4- Line 23: The brown planthopper (BPH), Nilaparvata lugens. Please consider throughout the text

Answer: We agreed and revised (Line 23).

Question 5- Line 24: Please move this sentence: We observed the tissue alterations in the host after infection by the pathogen using the morphological and histological methods” to the results section of the abstract

Answer: We agreed and moved with a minor revision of the expression (Lines 29-30).

Question  6- Line 28: Silver stain test (GMS). Filamentous fungi were first found on the external integument

Answer: We corrected and revised in Lines 29-30.

Question 7- Line 34: Please add a brief conclusion here.

Answer: We added in Lines 35-36.

“This study demonstrated that C. javanica PSUC002 is useful to control the BPH as eco-friendly practices and will possibly be applied in agriculture.”

Question 8- Line 55: Please provide the full scientific name of each species the first time.

Answer: We corrected and re-checked in Line 55-60.

Question 9- Line 66: by B. tabaci

Answer: We agreed and corrected in Line 69.

Question 10- Line 73: Please include a paragraph describing the cuticle-degrading enzymes' role.

Answer: We agreed and corrected in Lines 81-82.

Question 11- Line 76: DNA extraction and phylogenetic analysis of Cordyceps javanica

Answer: We agreed and corrected in Line 80.

Question 12- Line 80: reference please

Answer: We are sorry for the error and corrected in Line 84.

Question 13- Line 152: Measurement of cuticle-degrading enzymes requires a more detailed explanation.

Answer: We corrected Line 164-177.

Question 14- Line 187: The mortality rate of BPH-infected C. javanica

Answer: We agreed and corrected in Line 195.

Question 15- Line 252: Avoid starting a section with an abbreviation.

Answer: We agreed and re-checked.

Question 16- Line 252: This statement requires additional citations.

Answer: We corrected.

Question 17- Line 256: as a biocontrol agent of the

Answer: We corrected.

Question 18- The first paragraph of the discussion serves as the introduction.

Answer: We agreed and revised in 257-260, adding some introductory sentences.

Major concerns:

Question 19 - You truly mentioned that “Our observations confirmed that the activity of the sampled fungi peaked at 7 days of incubation”. By achieving this, what is your opinion on using this fungus for practical application in an IPM? Please discuss

Answer: We corrected and added in Line 297-303.

“Protease enzymes released from EPFs play a crucial role in degrading the insect cuticle [12-14]. The activity of the chitinase and protease sampled from C. javanica peaked at 7 days of incubation, which has an implication on the time course of killing BPHs. Although the precise enzymatic strategies of how C. javanica PSU002 kills the host are not well known, our results suggest that it is a week-long process. The production of EPF extracellular enzymes has a virulence potential for the pest insect [42]. The detailed time course should be investigated for the practical application”.

Question 20 - After a comprehensive analysis of the experiments and presentation of results, the Discussion section lacks depth. Thus, I suggest the authors enrich the discussion by scrutinizing the results, emphasizing possible explanations, and drawing comparisons.

Answer: Thank you very for your nice suggestion. We added several points to discussion, partly also by addressing other reviewers’ comments. We hope that our Discussion is improved.

Question 21- The statement above also applies to the conclusion.

Answer: We agreed and corrected in 314-315.

Reviewer 2 Report

Comments and Suggestions for Authors

The authors present a well rounded study of yet to be described entomopathogenic effects of C. javanica on BPH. While the scope of tests used are commendable, the manuscript could be improved considerably by revising organizational structures and enhancing explanations for why the study is important.

General comments:

·      The authors bury what I believe to be the most interesting finding, that BPH showed signs of decline before fungal infection was visible, within a poor organizational structure. Efficacy of a biocontrol agent primarily relies on how it influences host behavior in the field. Therefore, I would lead with findings about the host and then describe when fungal filaments and conidia were visible. Additionally, the abstract and basic summary do not mention mortality findings.

·      I think the authors need to address the misidentification of the entomopathogenic fungi more directly. The current intro and methods hop around between describing C. fumosorosea impacts to DNA identification without explanation. I believe a paragraph describing the difficulty identifying EPF in the field and what is known about similarity of impact among species within a shared genera or family would help the reader understand why the authors conducted DNA barcoding to identify the EPF. Additionally, reorganizing the methods to put identification last, or revising the section 2.3 to remove statements about how the EPF was misidentified originally would help the reader follow along with the study narrative.  As it’s currently written the authors seem apologetic about misidentifying the EPF, when really they can highlight that their study improves on other that may assume EPF identity without adequate molecular information. If the authors do not want to expand on the challenge of identifying EPFs as part of the study questions, I recommend all identification methods be removed to a supplement and that they should revise the text to simply say they investigated impacts of an untested EPF.

·      The authors take time to discuss results compared to other organisms (eg line 268-284) but do little in the introduction to set a reader up to anticipate why the sampe EPF could have varying effects within distinct hosts.

·      Overall this reads as a report of basic biocontrol efficacy tests rather than an interesting study of what makes an effective EPF and how we can use a variety of data to support a claim that an EPF could be effective in the field.

·      Using the active voice throughout the article would improve the overall clarity of ideas and tone of paper.

Line/section noted comments.

·      Simple summary needs to be revised entirely for active voice and overall clarity. It reads as a very rough first draft in comparison to the rest of the manuscript.

o   Line 14: Eg.  “We exposed brown planthoppers (BPH) Nilaparvata lugens, to the entomopathogenic fungus, Cordyceps javanica PSUC002 and observed morpho-histological changes 0 to 120 h post inoculation (pi). At 12 h pi we observed the first filamentous fungi on the external integument and the first fungal conidia penetrated host planthopper cells at 24 h pi.”

o   Make similar grammatical changes throughout simple summary.

o   Add a statement about why understanding the phases of leafhopper control by entomopathogenic fungi is important to help interpret the results.

·      Line 24: Add whether C. javanica is known to attack leafhoppers or BPH specifically.

·      Line 24-25: Combine lines 24-25 to avoid repeating information in the abstract

·      Line 29: Clarify to what the fungal conidia attached to?

·      Line 32-33: Combine the last two sentences of the abstract into one in which the metric of peak activity is clarified within a statement about when that peak activity occurred.

·      Abstract could do with a final sentence regarding the impact of these results on biocontrol outlooks for C. javanica on BPH

·      Line 122: First line of section 2.3 is repetitive from section 2.1

·      Figure 1. Panels B and C appear to convey redundant information. I believe Panel B displaying topological information is more relevant than panel C which appears to show major roadways and forested areas.

·      Section 2.5: Overall, this data analysis section needs to be revised to fully lay out the treatments tested. From how it’s currently written, it’s not clear what the Abbott’s formula is, or how it applies to mortality. It’s also not clear what treatments the ANOVA tested for (h pi?). Additionally, I think the statement about fungal area in BPH samples needs to go at the end of section 2.3 as it describes the measurement of infection impact not a statistical analysis.

·      Given Figure 8, Figure 5 & 6 needn’t to be in the body of the paper and may be better suited for a supplement.

Comments on the Quality of English Language

The basic summary needs to be revised for English clarity, but the overall manuscript reads quite well.

Author Response

Reviewer 2

Question 1 The authors present a well rounded study of yet to be described entomopathogenic effects of C. javanica on BPH. While the scope of tests used are commendable, the manuscript could be improved considerably by revising organizational structures and enhancing explanations for why the study is important.

Answer: We corrected and revised in the introduction such as Lines 70-74.

General comments:

Question 2 The authors bury what I believe to be the most interesting finding, that BPH showed signs of decline before fungal infection was visible, within a poor organizational structure. Efficacy of a biocontrol agent primarily relies on how it influences host behavior in the field. Therefore, I would lead with findings about the host and then describe when fungal filaments and conidia were visible. Additionally, the abstract and basic summary do not mention mortality findings.

Answer: We corrected and added in the mortality rate in Line 27. We also added the following parts to Discussion to emphasize this point.

Interestingly, a clear increase in BPH mortality was observed on Day 2, which was earlier than the occurrence of the necrosis. The fungal structures distributed within the body of BPH likely led to the death of the insect before inducing necrosis.

The enzymatic invasion to the body might be the cause of the mortality.

Question 2 I think the authors need to address the misidentification of the entomopathogenic fungi more directly. The current intro and methods hop around between describing C. fumosorosea impacts to DNA identification without explanation. I believe a paragraph describing the difficulty identifying EPF in the field and what is known about similarity of impact among species within a shared genera or family would help the reader understand why the authors conducted DNA barcoding to identify the EPF.

Additionally, reorganizing the methods to put identification last, or revising the section 2.3 to remove statements about how the EPF was misidentified originally would help the reader follow along with the study narrative.  As it’s currently written the authors seem apologetic about misidentifying the EPF, when really they can highlight that their study improves on other that may assume EPF identity without adequate molecular information. If the authors do not want to expand on the challenge of identifying EPFs as part of the study questions, I recommend all identification methods be removed to a supplement and that they should revise the text to simply say they investigated impacts of an untested EPF.

Answer: We thank you for your nice comments. We were afraid that adding a paragraph may be too much, but added a phrase in the last paragraph of Introduction (Line 79). We also added a phrase about the usefulness of DNA barcoding in Discussion “Thus, DNA barcoding is the only reliable method to correctly identify the species of this fungal group”.

With these corrections, we would like to keep the DNA data in the main text. We hope that it is now better fitted to the overall story of the paper.

Question 3 The authors take time to discuss results compared to other organisms (eg line 268-284) but do little in the introduction to set a reader up to anticipate why the sampe EPF could have varying effects within distinct hosts.

Answer:  We did not want to put too much information in Introduction but agree with the reviewer that the readers should be able to anticipate the species-specific nature of the EPF infection. We added the heads up in the 3rd paragraph of Introduction, which had some information about this issue.

Question 4 Overall this reads as a report of basic biocontrol efficacy tests rather than an interesting study of what makes an effective EPF and how we can use a variety of data to support a claim that an EPF could be effective in the field.

Answer: Thank you very much for your nice message. We plan to apply this EPF in the field study and modified the interpretation of the research emphasizing the practical point as suggested (Lines  315-316).

Question 5 Using the active voice throughout the article would improve the overall clarity of ideas and tone of paper.

Answer: We re-checked the manuscript and changed to the active voice to the degree that we felt natural.

Line/section noted comments.

Question 6 Simple summary needs to be revised entirely for active voice and overall clarity. It reads as a very rough first draft in comparison to the rest of the manuscript.

Line 14: Eg.  “We exposed brown planthoppers (BPH) Nilaparvata lugens, to the entomopathogenic fungus, Cordyceps javanica PSUC002 and observed morpho-histological changes 0 to 120 h post inoculation (pi). At 12 h pi we observed the first filamentous fungi on the external integument and the first fungal conidia penetrated host planthopper cells at 24 h pi.”

Answer: We re-checked and corrected.

Question 8 Make similar grammatical changes throughout simple summary.

Answer: Corrected in a similar way.

Question 9 Add a statement about why understanding the phases of leafhopper control by entomopathogenic fungi is important to help interpret the results.

Answer: We added the statement in Lines 34-36 and Lines 72-73.

Question10  Line 24: Add whether C. javanica is known to attack leafhoppers or BPH specifically.

Answer: The abstract was largely revised according to the comments from reviewers, but we incorporated this suggestion into the new version (Lines 24-25).

Question 11 Line 24-25: Combine lines 24-25 to avoid repeating information in the abstract

Answer: The abstract was largely revised according to the comments from reviewers, but we incorporated this suggestion into the new version (Lines 31-33).

Question 12 Line 29: Clarify to what the fungal conidia attached to?

Answer: We are sorry for the unclear description. We added “on the external integument at 12 h pi, and fungal conidia were attached to the integument at 24 h pi.”

Question 13 Line 32-33: Combine the last two sentences of the abstract into one in which the metric of peak activity is clarified within a statement about when that peak activity occurred.

Answer: The abstract was largely revised according to the comments from reviewers. We believe that the current version has good readability, but please let us known if the reviewer consideres that more revisions are needed.

Question 14 Abstract could do with a final sentence regarding the impact of these results on biocontrol outlooks for C. javanica on BPH

Answer: We agreed and added these sentence in line 34-36.

Question 15 Line 122: First line of section 2.3 is repetitive from section 2.1

Answer: We agreed and deleted the sentence.

Question 16 Figure 1. Panels B and C appear to convey redundant information. I believe Panel B displaying topological information is more relevant than panel C which appears to show major roadways and forested areas.

Answer: We agreed and corrected in Fig. 1.

Question 17 Section 2.5: Overall, this data analysis section needs to be revised to fully lay out the treatments tested. From how it’s currently written, it’s not clear what the Abbott’s formula is, or how it applies to mortality. It’s also not clear what treatments the ANOVA tested for (h pi?). Additionally, I think the statement about fungal area in BPH samples needs to go at the end of section 2.3 as it describes the measurement of infection impact not a statistical analysis.

Answer: We agreed and corrected the section as below.

Data analysis and statistical analysis

BPH mortality was corrected using Abbott’s formula and represented as a percentage (%) [29]. The mortality, fungal area and enzyme index were expressed as means ± SE. The mortality and fungal area were compared by ANOVA followed by Tukey’s HSD test. All data were analyzed by IBM SPSS Statistics 26.0 with a significance level of 95%.

Question 18 Given Figure 8, Figure 5 & 6 needn’t to be in the body of the paper and may be better suited for a supplement.

Answer: We understand that these figures convey redundant information (i.e., Fig. 8 is the summary figure). However, the journal Insects is an online journal that is generous page limitation. We hope to keep them to provide readers with detailed histological changes, which is the main finding of the present study.

Reviewer 3 Report

Comments and Suggestions for Authors

The effect of fungi of the genus Cordyceps on insects is a very interesting topic. Despite the good study of this issue, the authors of the article managed to conduct a series of experiments quite well and show new features of this process. The authors managed to get very interesting photographs. Unfortunately, the small size of these photographs will not allow readers to see the details of the Nilaparvata lugens infection process.

The article is of great scientific and practical importance, and after correcting technical shortcomings it can be recommended for publication.

Comments on individual elements of the article.

1. The words “the Brown Plant Hopper”, “(Stål)”, “PSUC002” should be removed from the title of the article.

2. Line 14, 23, 38: after the name of the species, you must indicate the surname and year.

3. Line 29-34: you need to add 4-5 sentences describing the processes in Figures 4, 5, 8.

4. Line 54, 55, 56, and others: in accordance with the International Code of Zoological Nomenclature, when first mentioned in the text of the article, the author’s surname and year must be added.

5. Line 81, 253 and others: if the sentence begins with an abbreviation of the generic name, in this case we do not abbreviate the generic name.

6. Line 117, 123: one degree of latitude is approximately 110 km. One millionth of a degree is 110 mm. It makes sense to indicate coordinates with an accuracy of up to 10 m, no more, that is, degrees must be rounded to the fourth decimal place.

7. Line 195: in the title you need to add “means ± SE” in parentheses. Also in this figure, you need to apply the correct method for comparing samples for each day of the experiment (for example, Tukey's test).

8. The size of figures 4, 5 and 8 should be increased by at least 40-50% in width and height. Each of them should take up a full page.

9. The letter designations above the columns in Figure 6 need to be carefully checked; the analysis is probably not carried out correctly.

10. In Figures 3 and 6 you need to write “Name of the abscissa or ordinate axis with a capital letter comma unit of measurement.”

11. On the ordinate axis of Figure 6, all numbers must be rounded to whole numbers.

12. Line 249: you need to add a repetition of the experiment. In each cell you need to write ± SE (not in parentheses). Above the numbers you need to put the letters a, b, c according to the results of the Tukey test.

13. In the Discussion, you need to add two large paragraphs about the biological features of Nilaparvata lugens and Cordyceps javanica known from the literature. Readers will be interested in this information.

14. Line 296-299: we need to add a broader view of the problem of using Cordyceps in plant protection. The link to the picture needs to be removed.

15. Line 320 and others: no need to capitalize all words.

16. Line 369, 400, 402 and others: italics and bold fonts are not highlighted correctly.

Author Response

Reviewer 3

Question 1. The words “the Brown Plant Hopper”, “(Stål)”, “PSUC002” should be removed from the title of the article.

Answer: We agreed and corrected.

Question 2. Line 14, 23, 38: after the name of the species, you must indicate the surname and year.

Answer: We agreed and corrected.

Question 3. Line 29-34: you need to add 4-5 sentences describing the processes in Figures 4, 5, 8.

Answer: We agreed and revised in Lines 30-31.

Question 4. Line 54, 55, 56, and others: in accordance with the International Code of Zoological Nomenclature, when first mentioned in the text of the article, the author’s surname and year must be added.

Answer: We agreed and corrected in Lines 55-60.

“especially Cordyceps fumosorosea (Hypocreales: Cordycipitaceae), formerly Isaria fumosorosea Wize, 1904, which has been used to control sap-sucking insects such as Jacobiasca formosana (Paoli, 1932) (Hemiptera: Cicadellidae), Aphis gossypii Glover, 187 (Hemiptera: Aphididae), Bemisia tabaci Gennadius, 1889 Gennadius (Hemiptera: Aleyrodidae) [8-9], and Stephanitis nashi Esaki & Takeya, 1931 (Hemiptera: Tingidae) [10].”

Question 5. Line 81, 253 and others: if the sentence begins with an abbreviation of the generic name, in this case we do not abbreviate the generic name.

Answer: We corrected and re-checked throughout the manuscript.

Question 6. Line 117, 123: one degree of latitude is approximately 110 km. One millionth of a degree is 110 mm. It makes sense to indicate coordinates with an accuracy of up to 10 m, no more, that is, degrees must be rounded to the fourth decimal place.

Answer: We agreed and corrected in Line 123.

Question 7. Line 195: in the title you need to add “means ± SE” in parentheses. Also in this figure, you need to apply the correct method for comparing samples for each day of the experiment (for example, Tukey's test).

Answer: We agreed and corrected.

Question 8. The size of figures 4, 5 and 8 should be increased by at least 40-50% in width and height. Each of them should take up a full page.

Answer: Thank you very much for your nice comments and we increased the figure size.

Question 9. The letter designations above the columns in Figure 6 need to be carefully checked; the analysis is probably not carried out correctly.

Answer: We corrected the letters and re-checking the statistics.

Question 10. In Figures 3 and 6 you need to write “Name of the abscissa or ordinate axis with a capital letter comma unit of measurement.”

Answer: We corrected and re-checked in Figs. 3 and 6.

Question 11. On the ordinate axis of Figure 6, all numbers must be rounded to whole numbers.

Answer: We corrected.

Question 12. Line 249: you need to add a repetition of the experiment. In each cell you need to write ± SE (not in parentheses). Above the numbers you need to put the letters a, b, c according to the results of the Tukey test.

Answer: We corrected as mean ± SE. However, the data in Table 1 was statistically tested (Line 254). We thank you for your nice comments.

Question 13. In the Discussion, you need to add two large paragraphs about the biological features of Nilaparvata lugens and Cordyceps javanica known from the literature. Readers will be interested in this information.

Answer: Thank you very much for your suggestion. Since our research is not focused on the biology of Nilaparvata lugens, we added the related description to Introduction as the background knowledge of this species (Lines 40-44). The biology of Cordyceps javanica was added to Discussion as suggested (Lines 270-277).

Question 14. Line 296-299: we need to add a broader view of the problem of using Cordyceps in plant protection. The link to the picture needs to be removed.

Answer: We corrected and deleted.

Question 15. Line 320 and others: no need to capitalize all words.

Answer: We agreed and corrected in Line 336.

Question 16. Line 369, 400, 402 and others: italics and bold fonts are not highlighted correctly.

Answer: We agreed and corrected.

Round 2

Reviewer 1 Report

Comments and Suggestions for Authors

I appreciate the authors for addressing my concerns. However, the discussion remains weak and requires further improvement.

Author Response

Editor and Reviewers

Thank you for all nice comments and help to increase the manuscript value of our research. All responses (II) are highlighted gray color, whereas the response I were showed as the red color.

All questions as you mentioned, and we are response below.

Question: I agree with the reviewers' comments. Enlarging photos is recommended.

Answer: We are sorry for this error and thank you for your nice comments.

In figures, they are enlarged in all figures and especially covered the pages 7 and 8. 

Question: Additional note: The authors demonstrated the proteolytic and chitinolytic activity of the fungus. They did not examine lipolytic activity. For this reason, the pragraph in the Discussion: “The fungi likely penetrated the integument using extracellular cuticle-degrading enzymes, which include proteolytic, chitinolytic, and lipolytic enzymes [12-14]. The present study detected the activity of these enzymes in C. javanica for the first time (Table 2)” is only partly justified.

Answer: We corrected and rechecked throughout the discussion.

Also, the re-wrote this data are corrected in Lines 292-297.